# Extracellular Traps Released by Neutrophils from Cats are Detrimental to *Toxoplasma gondii* Infectivity

**DOI:** 10.3390/microorganisms8111628

**Published:** 2020-10-22

**Authors:** Isabela S. Macedo, Marcos V. A. Lima, Jéssica S. Souza, Natalia C. Rochael, Pedro N. Caldas, Helene S. Barbosa, Flávio A. Lara, Elvira M. Saraiva, Rafael M. Mariante

**Affiliations:** 1Laboratório de Biologia Estrutural, Instituto Oswaldo Cruz, Fiocruz, Rio de Janeiro 21040-360, RJ, Brazil; macedos.isa@gmail.com (I.S.M.); marquinhos9796@gmail.com (M.V.A.L.); jsouza.biomed@gmail.com (J.S.S.); helene@ioc.fiocruz.br (H.S.B.); 2Laboratório de Imunobiologia das Leishmanioses, Instituto de Microbiologia Paulo de Góes, Universidade Federal do Rio de Janeiro, Rio de Janeiro 21941-902, RJ, Brazil; natyrochael@yahoo.com.br (N.C.R.); esaraiva@micro.ufrj.br (E.M.S.); 3HVN Hospital Veterinário Niterói, Niterói 24360-440, RJ, Brazil; pedrovet@pedrovet.com.br; 4Laboratório de Microbiologia Celular, Instituto Oswaldo Cruz, Fiocruz, Rio de Janeiro 21040-360, RJ, Brazil; flavioalveslara2000@gmail.com

**Keywords:** calcium, definitive host, elastase, felid, infectivity, killing, live imaging, neutrophil extracellular trap, PI3K, *Toxoplasma gondii*

## Abstract

*Toxoplasma gondii* is the causative agent of toxoplasmosis, an infectious disease that affects over 30% of the human world population, causing fatal infections in immunocompromised individuals and neonates. The life cycle of *T. gondii* is complex, and involves intermediate hosts (birds and mammals) and definitive hosts (felines, including domestic cats). The innate immune repertoire against the parasite involves the production of neutrophil extracellular traps (NET), and neutrophils from several intermediate hosts produce NET induced by *T. gondii*. However, the mechanisms underlying NET release in response to the parasite have been poorly explored. Therefore, the aims of this study were to investigate whether neutrophils from cats produce NET triggered by *T. gondii* and to understand the mechanisms thereby involved. Neutrophils from cats were stimulated with *T. gondii* tachyzoites and NET-derived DNA in the supernatant was quantified during the time. The presence of histone H1 and myeloperoxidase was detected by immunofluorescence. We observed that cat neutrophils produce both classical and rapid/early NET stimulated by *T. gondii*. Inhibition of elastase, intracellular calcium, and phosphatidylinositol 3-kinase (PI3K)-δ partially blocked classical NET release in response to the parasite. Electron microscopy revealed strands and networks of DNA in close contact or completely entrapping parasites. Live imaging showed that tachyzoites are killed by NET. We conclude that the production of NET is a conserved strategy to control infection by *T. gondii* amongst intermediate and definitive hosts.

## 1. Introduction

*Toxoplasma gondii* is an obligate intracellular protozoan parasite of the phylum Apicomplexa, and one of the most successful eukaryotic pathogens of medical and veterinary importance, since it can naturally infect all warm-blooded animals, including humans [1,2]. The parasite is distributed worldwide, and it is estimated that one third of the world’s human population is chronically infected with *T. gondii*, with a seroprevalence varying from 9% to more than 80% in different countries [3]. Humans acquire infection by ingesting contaminated food or water, consuming raw or undercooked meat containing the parasite, or by vertical transmission from mother to fetus [1,2]. In general, the clinical signs produced by *T. gondii* are mild and self-limiting in healthy individuals, but in some cases, it can lead to ocular toxoplasmosis [4]. On the other hand, in immunocompromised individuals and congenitally infected individuals the parasite can lead to severe illness [5,6].

The life cycle of *T. gondii* is complex, and the parasite can virtually infect all mammals and birds, that are intermediate hosts of the protozoan. Felids are the only known definitive host, in the gut of which the sexual cycle of the parasite occurs giving rise to the environmentally resistant oocysts [1,2]. The oocysts are released in cat feces and will mature in the environment, becoming a source of infection for other hosts [7]. Other forms of the parasite include the tachyzoites, which are the fast replicating forms that are vertically transmitted to fetus in congenital toxoplasmosis, and bradyzoites, which are present in tissue cysts and preferentially persist in brain and muscle tissues of infected hosts [8].

Neutrophils are the most abundant white blood cells in cats [9]. They constitute the first line of defense against invading pathogens and modulate adaptive immune responses by interacting with other immune cells through direct cell-to-cell contact [10] and the secretion of immunomodulators [11,12,13]. During *T. gondii* infection, neutrophils are rapidly recruited to the site of parasites entry [14,15,16], where they can get infected, therefore acting as motile reservoirs of *T. gondii* and contributing to the spread of the parasite through a Trojan-horse mechanism [16].

Neutrophils use an arsenal of weapons against invading microorganisms, including phagocytosis, degranulation, and production of extracellular traps [17,18,19,20]. Neutrophil extracellular traps (NET) are composed of a scaffold of DNA associated with granular and cytoplasmic proteins that can entrap and kill pathogens through two major mechanisms: a classical one, which is reactive oxygen species (ROS)-dependent and occurs after 1–4 h of stimulation, leading to neutrophil death; an early/rapid one, which does not depend on ROS and occurs within 5 to 15 min of stimulation, not affecting neutrophil viability and function [21,22,23].

Release of NET can be triggered by an uncountable number of biological stimuli, including bacteria, fungus, viruses, helminths, and protozoans such as the pathogenic parasite *T. gondii* [24,25,26,27,28]. Neutrophils from different organisms release NET upon interaction with *T. gondii*, including those of mice, humans, dogs, cattle, sheep, donkey, harbor seals, and dolphins [29,30,31,32,33,34,35], all of which are intermediate hosts for the parasite. Recently, extracellular traps were also shown to be produced by cat neutrophils [36]. However, the mechanisms behind NET formation in response to *T. gondii* are not completely understood.

Here, we show that neutrophils from cat, the main definitive host for the parasite, release NET in response to two of the major clonal lineages of *T. gondii* and that the parasites are entrapped and killed by NET. The release of NET begins as early as 15 min of interaction with parasites, and proceeds for several hours, suggesting the existence of both a rapid/early and a classical mechanism of NET production. Classical release of NET does not require infection of neutrophils, and does partially depend on elastase, intracellular calcium, and phosphatidylinositol 3-kinase (PI3K)-δ, but not on myeloperoxidase, peptidyl arginine deiminase (PAD)-4, and PI3Kγ.

Taken together, the data presented here allow us to speculate that the release of extracellular traps in response to *T. gondii* is a mechanism of the innate immunity to control infection conserved from intermediate to definitive hosts of the parasite.

## 2. Materials and Methods

### 2.1. Reagents

RPMI 1640 culture medium and fetal calf serum were obtained from Cultilab (Campinas, SP, Brazil). Ficoll Paque Plus was obtained from GE Healthcare (Chicago, IL, USA). DMEM/F12 culture medium was acquired from Gibco (Thermo Fisher Scientific, Waltham, MA, USA). Propidium iodide and penicillin-streptomycin antibiotics were obtained from Sigma-Aldrich (St. Louis, MO, USA). Quant-iT PicoGreen dsDNA Assay Kit, ProLong Diamond Antifade Mountant with DAPI, and PrestoBlue Cell Viability Reagent were acquired from Invitrogen (Thermo Fisher Scientific, Waltham, MA, USA).

We used the following pharmacological inhibitors/inducers in the study: phorbol 12-myristate 13-acetate (PMA; 100 nM; Sigma-Aldrich), DNase (20 U/mL; Invitrogen), cytochalasin D (CytD; 10 µg/mL; Sigma-Aldrich), elastase inhibitor III (ELi; MeOSuc-AAPV-CMK; 10 µg/mL; Calbiochem, San Diego, CA, USA), myeloperoxidase inhibitor I (MPOi; 600 nM; Calbiochem), AS605240 (selective inhibitor of PI3Kγ; 10 µM; Tocris Bioscience, Bristol, UK), IC87114 (selective inhibitor of PI3Kδ; 1 µM; Cayman Chemical, Ann Arbor, MI, USA), chloroamidine (Cl-A; PAD inhibitor; 12 μM; Cayman Chemical); PD98059 (MEK inhibitor; 60 µM; Sigma-Aldrich) and BAPTA/AM ([Ca^2+^]i chelator; 10 µM; Calbiochem).

The following antibodies were used in this study: rabbit anti-myeloperoxidase polyclonal (PA5-16672; 1:200; Invitrogen), mouse anti-histone H1 monoclonal (sc-8030; 1:100; Santa Cruz Biotechnology, Santa Cruz, CA, USA), mouse anti-toxoplasma (TP3) monoclonal (sc-52255; 1:100; Santa Cruz Biotechnology), Goat anti-Mouse Alexa Fluor 594 (A11032; 1:1600; Invitrogen), and Donkey anti-Rabbit Alexa Fluor 488 (A21206; 1:600; Invitrogen).

### 2.2. Animals

The cats used in this study were all mixed breed, from both sexes, aging from 1 to 6 years old, negative for feline immunodeficiency virus (FIV) and feline leukemia virus (FeLV). The study was carried out in strict accordance with the guidelines of the Oswaldo Cruz Foundation (Fiocruz), the National Council for the Control of Animal Experimentation (CONCEA, Brazil), and the Normative Resolution 30/2016 of the same entity. The blood samples herein used were leftovers from samples taken for the diagnose of animals in need.

### 2.3. Isolation of Neutrophils from Cats

Peripheral blood samples (≈2–4 mL) were provided by the Veterinary Hospital Niterói, RJ. Neutrophils were isolated by density gradient centrifugation using Ficoll Paque Plus (1.077 g/mL). After centrifugation (400× *g* for 30 min), the neutrophil layer containing red blood cells (RBC) was collected and transferred to a 15 mL conical tube and suspended in 0.2% NaCl for 30 s for RBC lysis. Afterwards, an equal amount of NaCl 1.6% was added to stop RBC lysis. Finally, cells were washed, suspended in RPMI 1640 medium supplemented with 1% fetal calf serum and 1% penicillin-streptomycin antibiotics solution, and kept on ice until use.

### 2.4. Parasites Culture

Tachyzoites of *T. gondii* from RH strain are routinely maintained in Swiss mice by intraperitoneal inoculation of parasites, as previously described [37]. After 72 h of inoculation parasites were harvested from peritoneum in PBS and centrifuged (200× *g* for 5 min) to remove any contaminating cells. The supernatant containing the parasites was transferred to another tube and centrifuged again (1500× *g* for 10 min). Parasites were recovered from pellet, counted in a hemocytometer, and used to infect subconfluent monolayers of Vero cells (Vero ATCC CCL-81™, Manassas, VA, USA) or to stimulate NET production, as described later. Vero cells cultures re maintained at 37 °C, 5% CO_2_ in DMEM/F12 culture medium supplemented with 10% fetal calf serum and 1% penicillin-streptomycin antibiotic solution. Tachyzoites released from the supernatant of infected Vero cells a few days later were also used to stimulate NET production. They were harvested and isolated by differential centrifugation exactly as explained above for the peritoneum. To maintain their virulence, parasites are periodically passed in vivo.

Parasites of *T. gondii* ME49 strain are routinely maintained in C57BL/6 mice. Tissue cysts are obtained from the brains of mice inoculated intraperitoneally with 50 cysts/animal, as described elsewhere [38]. Briefly, cysts were ruptured with acid pepsin solution and free parasites were added to Vero cells cultures. Two weeks later, released tachyzoites from infected cells were collected from the supernatant as described above for the RH strain, washed, counted in a hemocytometer, and used to infect other Vero cultures or to stimulate NET production. To maintain their virulence, parasites are periodically passed in vivo.

*Leishmania amazonensis* promastigotes from MHOM/BR/77/LTB0016 strain were maintained at 26 °C in Schneider’s insect medium supplemented with 10% fetal calf serum. Stationary-phase promastigotes were obtained from 5-day-old cultures. Parasites were washed and counted in a hemocytometer.

### 2.5. NET Induction/Inhibition Assay

In order to induce NET production, neutrophils (10^5^ cells per well) were plated in 96-well plates and incubated with PMA, *L. amazonensis* promastigotes or *T. gondii* tachyzoites from either RH or ME49 strains (5:1 parasites/neutrophil ratio) for up to 180 min at 37 °C with 5% CO_2_. Paraformaldehyde-fixed tachyzoites were used at the same parasite/neutrophil ratio as well. In some cases, neutrophils were treated for 30 min with one of the several pharmacological inhibitors listed above before adding *T. gondii* tachyzoites. Afterwards supernatants were collected, spun down for 5 min, and stored at −80 °C until use.

### 2.6. Quantification of NET-Derived DNA

Supernatants were distributed into 96-well opaque plates, and NET-derived DNA was quantified using the Quant-iT PicoGreen dsDNA Assay Kit according to manufacturer’s instructions. Analysis was performed in a SpectraMax Paradigm microplate reader (Molecular Devices, Sunnyvale, CA, USA) using 485/538 nm excitation/emission wavelengths. Herring sperm DNA was used to make the standard concentration curve.

### 2.7. Detection of Myeloperoxidase, Histone H1, and T. gondii

Neutrophils (2.5 × 10^5^ cells per well) were plated in 24-well plates with coverslips and incubated or not with *T. gondii* tachyzoites from RH strain for 180 min at 37 °C, 5% CO_2_. Cultures were fixed with 4% paraformaldehyde for 30 min at room temperature and carefully washed with PBS. Coverslips were stained with anti-MPO and either anti-Histone H1 or anti-*Toxoplasma gondii* antibodies, followed by the secondary antibodies anti-mouse Alexa Fluor 594 and anti-rabbit Alexa Fluor 488. Slides were mounted with ProLong Diamond Antifade Mountant with DAPI and examined and photographed in a Zeiss Axio Imager.M1 microscope (Zeiss, Germany).

### 2.8. Ultrastructural Visualization of T. gondii Entrapment in NET

Neutrophils (2.5 × 10^5^ cells per well) were cultivated in 24-well plates with coverslips and stimulated with *T. gondii* tachyzoites from RH strain for 180 min at 37 °C, 5% CO_2_. Cells were fixed for 1 h at room temperature with 2.5% glutaraldehyde in 0.1 M sodium cacodylate buffer (pH 7.2) containing 3.5% sucrose and 2.5 mM CaCl_2_. After washing, cells were post-fixed with 1% osmium tetroxide and 0.8% potassium ferricyanide for 1 h at room temperature and dehydrated in crescent concentrations of ethanol. Following dehydration, the specimens were critical-point dried, coated with gold, and observed in a JEOL-JSM-6390LV scanning electron microscope (JEOL, Japan) from the Rudolf Barth Electron Microscopy Platform (Instituto Oswaldo Cruz, Fiocruz, Rio de Janeiro, RJ, Brazil).

### 2.9. Production of NET-Enriched Supernatants

Neutrophils (6 × 10^5^ cells per well) were plated in 24-well plates and incubated with *T. gondii* tachyzoites from RH strain (5:1 parasites/neutrophil ratio) for 180 min at 37 °C, 5% CO_2_. Afterwards supernatant was collected, spun down for 5 min, and stored at −80 °C until use. NET-DNA was quantified with the Quant-iT PicoGreen dsDNA Assay Kit as described before. NET-enriched supernatants were treated or not with DNase (20 U/mL) for 30 min and used in the parasite infectivity assay.

### 2.10. Parasite Infectivity Assay

*T. gondii* tachyzoites from RH strain were incubated in NET-enriched supernatants (≈1 µg/mL NET-DNA) for 180 min at 37 °C, 5% CO_2_. Supernatants were pretreated or not with 20 U/mL DNase for 30 min before adding parasites. Afterwards parasites were used to infect monolayers of Vero cells (1:1 parasite/Vero ratio) in coverslips for 180 min. Cells were then washed to remove non-internalized parasites and kept at 37 °C, 5% CO_2_, for 21 h. Coverslips were stained with Giemsa and observed in a Zeiss Axio Imager.A2 microscope. At least 200 cells were counted randomly in the central area of each slide. The infection index was calculated using the following formula:$$\mathrm{Infection}\;\mathrm{Index}=\frac{\%\;\mathrm{Infected}\;\mathrm{Cells}\times \mathrm{Total}\;\mathrm{Number}\;\mathrm{of}\;\mathrm{Intracellular}\;\mathrm{Parasites}}{\mathrm{Total}\;\mathrm{Number}\;\mathrm{of}\;\mathrm{Cells}}$$

### 2.11. Neutrophil Viability Assay

The cytotoxicity of the inhibitors to neutrophils was examined with PrestoBlue Cell Viability Reagent according to manufacturer’s instructions. Briefly, neutrophils were incubated with inhibitors in the same conditions used for the NET induction assay, except that cells were not further stimulated. Cells were kept at 37 °C, 5% CO_2_, for 180 min, and PrestoBlue was added 20 min before the end of the incubation time. Analysis was performed in a SpectraMax M2 microplate reader (Molecular Devices) using 560/590 nm excitation/emission wavelengths. Data is represented as percentage of control.

### 2.12. Live Cell Imaging

Neutrophils (5 × 10^5^) were seeded in 35 mm CELLview plates (Greiner Bio-One, Americana, SP, Brazil) and allowed to adhere for 30 min at 37 °C. Afterwards, non-adherent cells were washed out and *T. gondii* tachyzoites from RH strain were added at a parasites:neutrophil ratio of 5:1 in 800 µL of RPMI. Propidium iodide was added to the dish at a final concentration of 3 µg/mL, in order to stain NET as well as dead neutrophils and parasites over time. Parasites were allowed to settle for 15 min before recording began. Spontaneous death of parasites in culture medium in the absence of neutrophils was also evaluated as a control condition. Time-lapse analysis was performed by sequential acquisition along several hours with a time interval of 30 s, using a Zeiss Axio Observer Z1 microscope. Images were acquired by an HMR Axiocam monochrome camera operated by Axiovision software version 3.2 (Zeiss, Germany). The red signal from propidium iodide was acquired by Colibri illumination system using a 590 nm LED with Zeiss fluorescence filter 50. Temperature and focus were maintained along time-lapse analysis using a Temperature Control module and a Zeiss Definitive Focus device, respectively. Images were processed in ImageJ 1.52 (National Institutes of Health, Bethesda, MD, USA).

### 2.13. Statistical Data Analysis

The experiments were generally performed in duplicate and repeated two to six times, according to the number of blood samples (number of cats) available at the moment and the number of neutrophils that we were able to obtain from each blood sample. The number of cats for each day of experimentation ranged from one to six. Data are presented as the total number of cats tested for each experiment as mean  ± SD values. Comparisons between groups were done by paired t-test, after D’Agostino–Pearson omnibus normality test, or repeated measures one-way ANOVA with Greenhouse–Geisser correction and Dunnett’s multiple comparisons test, when appropriate. Differences of *p* <  0.05 were considered to be significant. GraphPad Prism 7 software (GraphPad, San Diego, CA, USA) was used for all analyses.

## 3. Results

### 3.1. Neutrophils from Cats Produce Extracellular Traps in Response to Classical Inducers

We first investigated the ability of feline neutrophils to produce NET in response to either PMA, a well-known PKC activator and potent inducer of NET [39], or *L. amazonensis*, a protozoan parasite with a well-described capacity to induce NET both in vitro and in vivo [40,41,42], as our positive control conditions. As expected, treatment with PMA led to a two-fold increase in dsDNA release when compared to untreated cells (Appendix A). Stimulation of cells with *L. amazonensis* promastigotes induced a seven-fold increase in dsDNA production as compared to control cells (Appendix A), corroborating our previous findings [43,44]. These results confirm that neutrophils from cats are competent in producing NET triggered by various stimuli as part of its repertoire of immune responses.

### 3.2. Cat Neutrophils Produce Extracellular Traps in Response to Toxoplasma gondii

We next examined whether neutrophils from cats produce NET as a strategy to combat infection by *T. gondii*. NET production was accompanied for up to 180 min after incubating neutrophils with tachyzoites from two clonal strains of the parasite, RH (type I, virulent strain) or ME49 (type II, cystogenic strain). *T. gondii* induced dsDNA release by neutrophils in a strain- and time-dependent manner, beginning as early as 15 min after incubation with parasites (Figure 1). Interestingly, production of NET in response to RH strain was more pronounced than to ME49 strain at later time points (Figure 1A,E; 4.0- and 4.9-fold change for RH and 2.9- and 2.7-fold change for ME49 at 60 and 180 min, respectively, when compared to control). This assay was performed with neutrophils from at least 10 different cats, and although the extent of the response varies from one donor to the other, the response pattern is the same for all donors (Figure 1B,F). Paraformaldehyde fixed parasites induced far less NET than live parasites (Figure 1C,G), suggesting that NET production may have the contribution of factors released by *T. gondii*. We cannot rule out the possibility that the fixation process itself could alter the conformation of the parasite’s surface molecules [45], thereby affecting their recognition by the receptors involved in NET release.

In order to determine if NET production in response to live parasites would be triggered during invasion of neutrophils by *T. gondii*, we treated neutrophils for 30 min with the actin polymerization inhibitor cytochalasin D before adding the parasites. Pretreatment of cells with CytD did not affect dsDNA release by cat neutrophils (Figure 1D,H), indicating that the invasion of cells is dispensable for NET production in response to *T. gondii*.

### 3.3. NET from Cats Entrap Toxoplasma gondii Tachyzoites

In order to investigate the components associated with feline NET and the capacity of those NET to entrap parasites, we performed a microscopic analysis of the parasite–NET interaction. After 180 min interaction of *T. gondii* with neutrophils, cells were fixed, stained for myeloperoxidase, histone H1 or the parasite and DNA, and examined in a fluorescence microscope. We found that NET released in response to *T. gondii* tachyzoites contains all the classical NET signatures (Figure 2A), and that many parasites are entangled in DNA-MPO structures (Figure 2B).

We further analyzed the ultrastructural aspects of parasite–NET interactions. Neutrophils were incubated with *T. gondii* tachyzoites for 180 min, fixed, processed as described in methods, and observed in a scanning electron microscope. Three representative fields were photographed at different magnifications, showing either strands or networks of extracellular DNA in close association with *T. gondii* tachyzoites (Figure 3). In some cases, parasites can be seen completely entrapped in the network of NET produced by cat neutrophils (Figure 3G,J,K).

### 3.4. Release of dsDNA from Cat Neutrophils in Response to T. gondii Depends on Elastase, Calcium, and PI3Kδ

To further elucidate the mechanisms involved in NET release by cat neutrophils in response to *T. gondii* tachyzoites, we pretreated neutrophils with one of several inhibitors before adding parasites. We first assessed the role of elastase, myeloperoxidase, and PAD4 on dsDNA release after 180 min. Administration of elastase inhibitor decreased NET-derived DNA production by cat neutrophil by 27% (Figure 4A). MPO or PAD inhibition did not affect dsDNA release by neutrophils stimulated with *T. gondii* (Appendix A). Importantly, the inhibitors herein used were not toxic to neutrophils (Appendix A).

Next, we investigated the role of calcium, PI3K, and MEK signaling pathways in feline NET induced by *T. gondii*. Pretreatment of neutrophils with a calcium chelator led to a 22% inhibition in dsDNA release by cat neutrophils (Figure 4B). Administration of a PI3Kδ selective inhibitor decreased by 26% the release of NET-derived DNA triggered by *T. gondii* (Figure 4C). Interestingly, treatment of feline neutrophils with a PI3Kγ selective inhibitor had no effect on dsDNA release in response to tachyzoites (Appendix A). Inhibition of MEK had no effect on dsDNA release by cat neutrophils in the presence of *T. gondii* (Appendix A). Of note, the calcium chelator and the inhibitors for PI3Kδ, PI3Kγ, and MEK presented no toxicity to neutrophils after 180 min (Appendix A).

### 3.5. NET from Cats Kill Toxoplasma gondii Tachyzoites

To verify the capacity of feline NET to affect infectivity of *T. gondii*, we incubated parasites for 180 min in NET-enriched supernatants pretreated or not with DNase and examined the ability of the tachyzoites to infect Vero cells. We found an infection index 47% higher when Vero cells were incubated with parasites treated with NET-enriched supernatants in the presence of DNase than with parasites treated in the absence of DNase (Figure 5), suggesting that NET directly inhibit *T. gondii* infectivity.

Finally, we asked whether inhibition of *T. gondii* infectivity could be due to parasite killing by NET. We performed a time-lapse analysis of neutrophils incubated with *T. gondii*, and found that neutrophils can release NET from as early as 15 min to up to 180 min of interaction with parasites (Figure 6 and Appendix A depict the release of NET after 80 min interaction with the parasite). As expected, parasites entrapped in NET died over time, as seen by propidium iodide incorporation (Figure 7), suggesting that loss of host cells infectivity is caused by a loss of *T. gondii* tachyzoites viability. Noteworthy, parasites incubated in culture medium in the absence of neutrophils remained alive for up to 5 h (Appendix A), confirming that the death of tachyzoites was caused by NET.

## 4. Discussion

Although several studies have called attention to the importance of neutrophils in the infection and spread by *Toxoplasma gondii* in mice [16,29,46,47,48], the role of neutrophils in feline toxoplasmosis remains poorly explored. In general, gaining knowledge in the field of parasite–neutrophil interaction will help to elucidate the mechanisms involved in the evasion of the parasite from the immune system and in the establishment of the disease. Here, we elucidated part of the molecular mechanisms involved in NET production induced by *T. gondii* in cat neutrophils, evidencing the participation of neutrophil elastase, intracellular calcium, and PI3Kδ in the process.

Neutrophil extracellular trap production in response to *T. gondii* was first described by Abi Abdallah and colleagues, showing that neutrophils from humans and mice release NET induced by tachyzoites through a MAPK-partially dependent pathway, and that NET reduce parasite viability [29]. A few years later, production of NET in response to *T. gondii* infection was shown for other intermediate hosts. Neutrophils from dog [35], sheep and cattle [31], donkey [34], harbour seal [30], and dolphin [33] all release NET induced by this parasite. Moreover, Lacerda and colleagues recently showed that neutrophils from cats also produce NET in response to *T. gondii* [36], although without exploring its mechanism or NET toxicity to the parasite. Here we show that cat neutrophils produce classical NET in response to tachyzoites of the virulent strain RH, corroborating previous findings [36]. Furthermore, we show that the cystogenic strain ME49 also induces NET release, and that production of NET by cat neutrophils in response to both virulent or cystogenic strains occurs either in a classical way, after a few hours, or in a rapid way, within the first 15 min of interaction with parasites. Early NET release has been previously shown for neutrophils from intermediate hosts in response to RH strain [30,31,34,35] and from bovine neutrophils stimulated with a related apicomplexan parasite, *Besnoitia besnoiti* [49]. Zhou and colleagues present evidence that bovine neutrophils produce NET within ≈30 min of interaction with *B. besnoiti* bradyzoites, and that the overall phenotype and crawling activities of neutrophils seem not to be affected by this event, suggesting that the rapid NET release is vital [49]. Those results corroborate previous findings with unrelated protozoan [23] and other microorganisms [22]. Whether neutrophils from cats remain alive after rapid NET release in response to *T. gondii* deserves further investigations.

We found that production of cat NET in response to the virulent RH strain was more pronounced than that triggered by the cystogenic ME49 strain over time. These differences might somehow reflect the differences in host immune response induced by those strains [50]. Moreover, it seems that the parasites affect NET production by releasing soluble factors, and not only by direct contact, since paraformaldehyde fixed tachyzoites induced much less NET than live ones. In this line of reasoning, Abi Abdallah and colleagues showed that internalization of tachyzoites by human neutrophils is not necessary for NET production [29]. Here we corroborate those findings and show that pretreatment of cat neutrophils with cytochalasin—a drug that inhibits actin polymerization, thereby preventing phagocytosis—does not significantly impact NET production. However, we cannot rule out the possibility that the phagocytosis-independent active internalization of parasites, especially in the virulent RH strain [51,52], can play a role in the process of NET production. In addition, the fixation of tachyzoites with paraformaldehyde could alter the conformation of molecules [45] on the parasite’s cell surface, which could somehow be involved in neutrophil receptor recognition during NET release. Additional studies are necessary to elucidate these questions.

Further, we examined the components and ultrastructural aspects of cat NET and its capacity to ensnare tachyzoites of the virulent RH strain of *T. gondii*. We identified histone H1 as well as myeloperoxidase associated with the released DNA. Moreover, several tachyzoites were seen associated with cat NET. By scanning electron microscopy, we observed strands as well as networks of chromatin fibers entangling parasites. Previous studies have categorized NET according to different morphological structures: diffuse NET (*diff*NET), consisting of extracellular decondensed chromatin presented in a circular form and close to the cell body; spread NET (*spr*NET), corresponding to elongated web-like structures composed of thin chromatin fibers; aggregated NET (*agg*NET), characterized by large clusters of NET aggregating many neutrophils with a massive appearance [53,54,55]. We show here that the NET produced by cat neutrophils after 180 min of interaction with RH tachyzoites are very similar to *spr*NET. Imlau and colleagues show that neutrophils isolated from dolphins produce all three types of NET, with a predominance of *spr*NET and *agg*NET [32]. Here, we did not find large clusters of NET aggregating neutrophils, but eventually small clusters could be seen (Appendix A), corroborating previous findings and suggesting a conserved mechanism of NET release in response to *T. gondii*.

NET’s antimicrobial activity can lead to death of various groups of pathogens, including bacteria, fungi, and protozoa. This is accomplished by the presence in NET of a plethora of proteases, antimicrobial enzymes, histones, and DNA, all of which contribute toward direct antimicrobial effects [56]. Here we investigated the ability of cat NET to inhibit infectivity and kill ensnared *T. gondii*. We show that treatment of NET-enriched supernatants with DNase increases the infectivity of tachyzoites to epithelial cells, and that entangled parasites are killed by NET, corroborating previous findings with mice neutrophils [29]. Interestingly, our live cell imaging analysis reveals that not all the parasites associated with NET died over a long time frame of recording. There is growing evidence that some microorganisms developed the ability to escape NET by degrading NET structure, inhibiting NET release, or resisting NET killing [57]. Degradation of NET are usually achieved by nucleases produced by pathogens, and we have previously shown that the expression of the enzyme 3′-nucleotidase/nuclease (3′NT/NU) by *Leishmania* parasites contributes to their escape from NET killing [58]. On the other hand, nuclease production can be associated with the degree of virulence of some parasites [57,59]. Therefore, it is tempting to speculate that the virulent RH tachyzoites we used in our assays are producing nucleases or other factors which could degrade NET and enable the survival and/or escape of at least part of the parasite population. Corroborating this hypothesis, Wei and colleagues recently showed that *T. gondii* tachyzoites from the same strain can degrade NET induced by zimosan [35]. Additional investigations are needed to elucidate the molecular factors involved there.

The mechanisms involved in the release of NET have been focus of intensive investigation in recent years. Although not fully understood, the molecular mechanisms of classical NET release involves different molecules and signaling pathways, including the activation of the Raf-MEK-ERK pathway and the generation of ROS by NADPH oxidase [53,60], the translocation of neutrophil elastase to the nucleus, and the association of MPO to the DNA, leading to chromatin decondensation [61] and the citrullination of histones by the nuclear enzyme peptidyl arginine deiminase-4 (PAD4), mediated by intracellular Ca^2+^ influx and ROS, also leading to decondensation of chromatin [62,63]. Reichel and colleagues showed that inhibition of neutrophil elastase or MPO completely abrogated dsDNA release by neutrophils from harbour seal stimulated with *T. gondii* [30]. However, it is not clear whether NET production by cat neutrophils depends on MPO or elastase. We therefore examined the role of these molecules and found that inhibition of elastase but not of MPO significantly affected dsDNA release, indicating that elastase is more important than MPO in our model.

In a recent report, Wei and colleagues showed that treatment of dog neutrophils with the NADPH oxidase inhibitor diphenylene iodonium (DPI) prevented DNA release in response to *T. gondii* [35]. Similar results were obtained with donkey neutrophils [34]. In another study, Reichel and others disclosed that inhibition of NADPH oxidase with the same inhibitor prevented NET release by neutrophils from harbour seal stimulated with *T. gondii*. Since the mechanism of ROS production is dependent on Ca^2+^ and the store-operated Ca^2+^ entry (SOCE) plays a fundamental role in this process [64,65], the authors investigated the involvement of Ca^2+^ and showed that the inhibition of the SOCE pathway significantly reduced dsDNA release, suggesting the participation of Ca^2+^/SOCE in the process of NET release in harbour seal [30]. Recently, cat neutrophils have been shown to produce ROS after stimulation with *T. gondii* [36]. However, if the release of NET by cat neutrophils depends on the production of ROS is something that deserves further investigation. Unfortunately, we could not provide additional information about the direct role of NADPH oxidase and ROS production in the process of NET release by cat neutrophils, since treatment of our cells with DPI led to massive neutrophils death (data not shown). On the other hand, we show the involvement of Ca^2+^ in cat NET production after treating cells with the intracellular Ca^2+^ chelator BAPTA/AM, corroborating previous findings [30]. Interestingly, treatment with the PAD inhibitor chloroamidine did not prevent NET release by cat neutrophils, suggesting the existence of a PAD-independent mechanism of NET release that might be influenced by intracellular Ca^2+^ concentration.

We have shown previously that production of NET induced by *Leishmania amazonensis* is partially dependent on PI3K signaling pathway, either through PI3Kδ and PI3Kγ isoforms [44]. These isoforms are commonly expressed in leukocytes, and together they regulate several functions of the immune system [66]. Whereas PI3Kγ regulate ROS generation, PI3Kδ is dispensable for this process, although it is important for the amplification of a second wave of ROS production dependent on the first wave initiated by PI3Kγ [67]. Here, we showed that the blockage of PI3Kδ with a selective inhibitor decreased NET release in response to *T. gondii* by cat neutrophils. Surprisingly, administration of a PI3Kγ selective inhibitor had no effect on NET production in our model, indicating that generation of ROS might be secondary in NET release by cat neutrophils. Accordingly, MEK1/2 inhibition also had no effect on NET release by cat neutrophils in the presence of *T. gondii*, unlike what was found for human neutrophils, where NET production involves activation of ERK1/2 signaling pathway and where blockage of MEK1/2 partially prevents the release of NET [29]. These discrepancies may reflect the existence of different signaling pathways for NET release between different species in response to the *T. gondii*.

Much progress has been made in recent years with regard to the need of different molecular pathways to induce NET depending on the stimulus, and despite being important for the classical NET release for some stimuli, NAPDH oxidase, ROS, MPO, and neutrophil elastase are not necessary to induce NET by other stimuli [39,68]. Therefore, we speculate here that although production of NET is a conserved mechanism between different species, neutrophils from different organisms may use different signaling pathways to induce NET to the same stimulus. Further comparative studies are needed to test this hypothesis.

## 5. Conclusions

We show for the first time that neutrophils from cats release both classical and rapid/early NET in response to either virulent or cystogenic strains of *T. gondii*, and that the parasites are entrapped and killed by NET. The mechanisms of classical release of cat NET to *T. gondii* tachyzoites involve the participation of neutrophil elastase, intracellular calcium, and PI3Kδ signaling pathways.

## Figures and Tables

**Figure 1 microorganisms-08-01628-f001:**
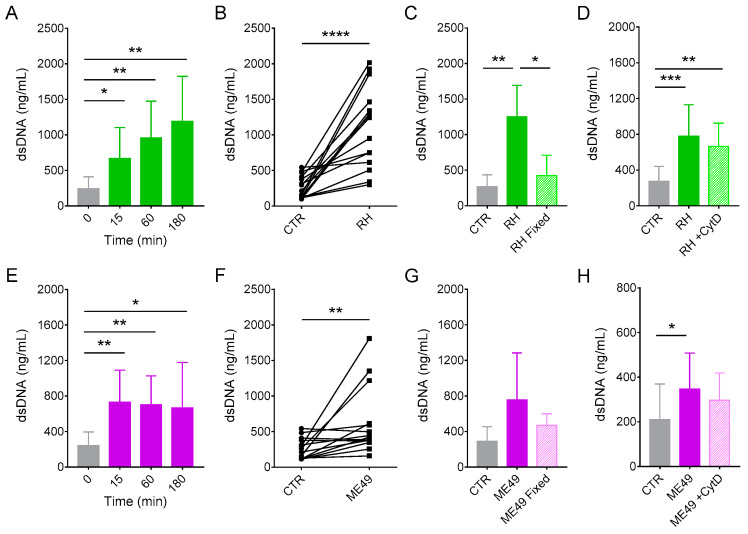
*Toxoplasma gondii* induce neutrophil extracellular trap (NET) release from cat neutrophils. Neutrophils from several donors were incubated for the indicated time intervals in the absence (CTR) or presence of tachyzoites from (**A**—**D**) RH or (**E**—**H**) ME49 strains (5:1 parasites:neutrophil ratio). Supernatants were collected and released dsDNA was quantified with PicoGreen Kit. (**A**,**E**) Release of NET from feline neutrophils is strain- and time-dependent (*n* = 10–12). (**B**,**F**) Interdonor variations in dsDNA release after 180 min incubation with parasites (*n* = 14–15). (**C**,**G**) NET induction by paraformaldehyde-fixed or live parasites after 180 min incubation (*n* = 8–9). (**D**,**H**) Formation of NET does not require invasion of neutrophils by the parasites. Cat neutrophils were pretreated or not with cytochalasin D (CytD; 10 µg/mL) for 30 min and then stimulated with *T. gondii* tachyzoites (*n* = 5–12). All results are shown as mean (SD). * *p* < 0.05; ** *p* < 0.01; *** *p* < 0.001; **** *p* < 0.0001.

**Figure 2 microorganisms-08-01628-f002:**
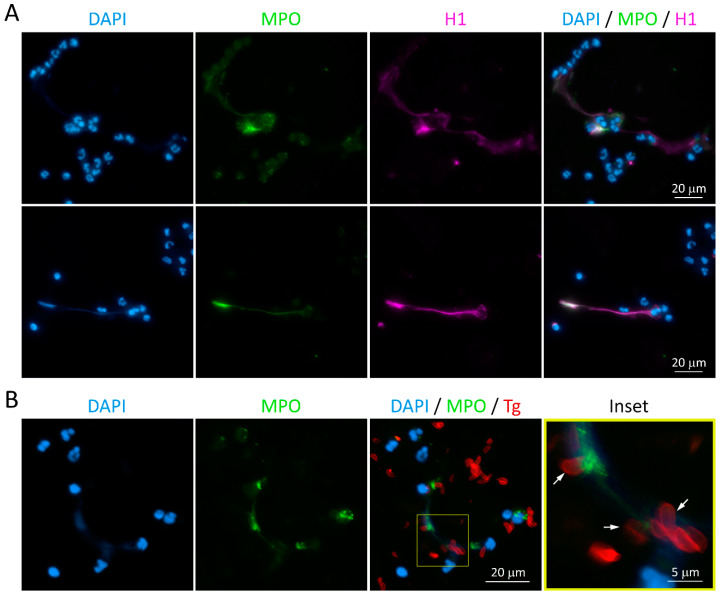
NET formation induced by *Toxoplasma gondii* tachyzoites. Neutrophils were incubated for 180 min with RH strain tachyzoites (5:1 parasites:neutrophil ratio), fixed and stained for myeloperoxidase (MPO) and histone H1 (**A**) or *T. gondii* (**B**). Neutrophils DNA was counterstained with DAPI. Inset shows a region containing several parasites entrapped in DNA-MPO rich structures (arrows). Tg = *T. gondii*.

**Figure 3 microorganisms-08-01628-f003:**
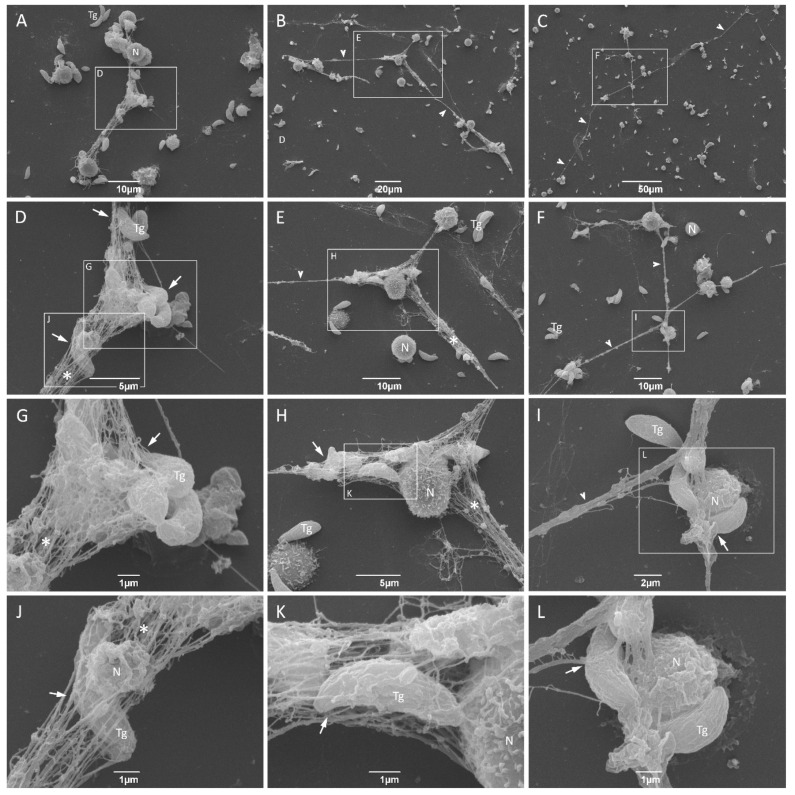
Ultrastructural aspects of *T. gondii* and cat NET interactions. Neutrophils were incubated for 180 min with tachyzoites from RH strain (5:1 parasites:neutrophil ratio), fixed and processed for scanning electron microscopy (**A**–**L**). Strands (arrowheads) or networks (asterisks) of NET can be seen entrapping tachyzoites of *T. gondii* (arrows). Selected areas indicated by rectangles are zoomed in the picture bellow. Tg = *T. gondii*; N = neutrophil.

**Figure 4 microorganisms-08-01628-f004:**
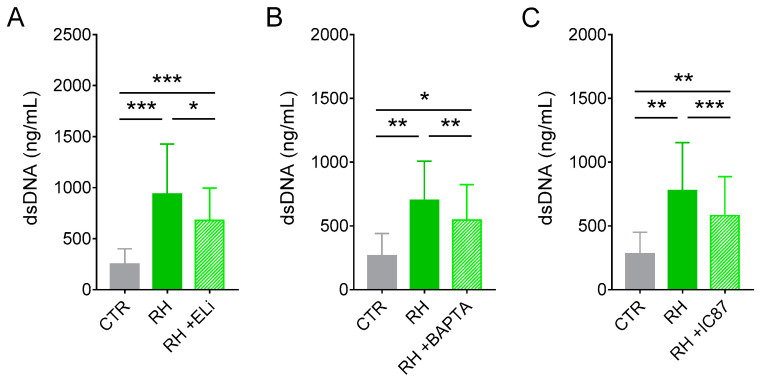
Classical NET-derived DNA release in response to *T. gondii* involves elastase, intracellular calcium, and PI3Kδ. Neutrophils were pretreated or not with (**A**) the elastase inhibitor MeOSuc-AAPV-CMK (Eli; 10 µg/mL), (**B**) the intracellular calcium chelator BAPTA/AM (BAPTA; 10 µM), or (**C**) the PI3Kδ selective inhibitor IC87114 (IC87; 1 µM) for 30 min and then stimulated for 180 min with RH tachyzoites (5:1 parasites:neutrophil ratio). Supernatants were collected and released dsDNA was quantified with PicoGreen Kit (*n* = 9–13). All results are shown as mean (SD). * *p* < 0.05; ** *p* < 0.01; *** *p* < 0.001.

**Figure 5 microorganisms-08-01628-f005:**
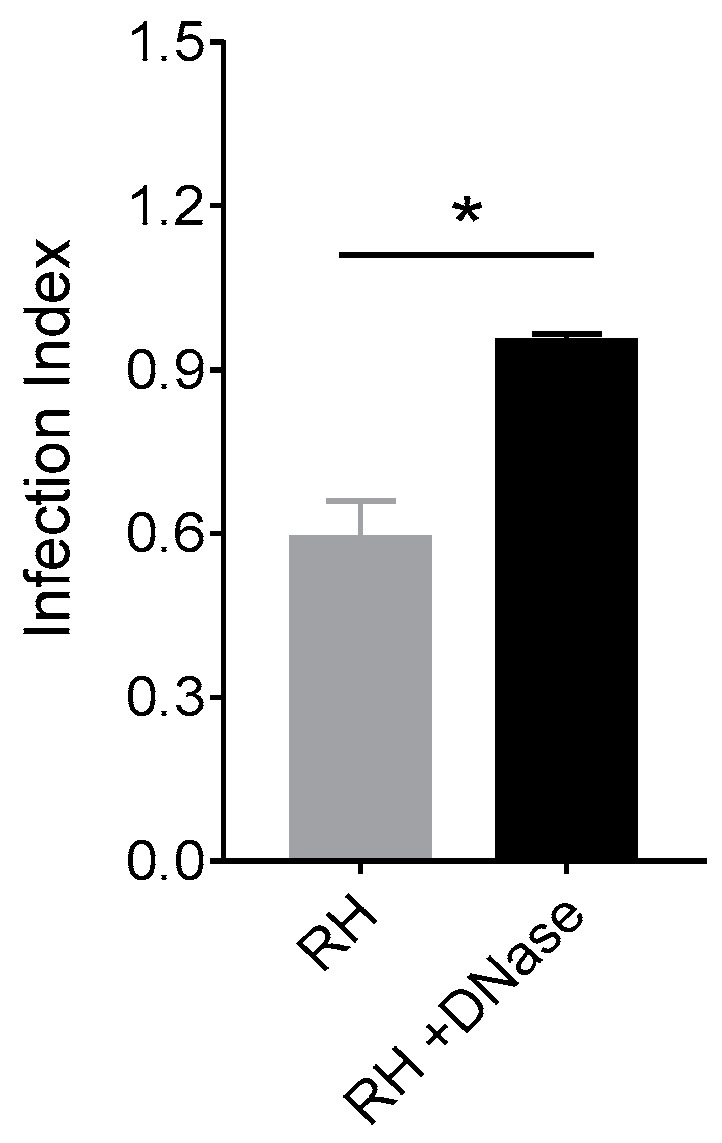
NET decreases *T. gondii* infectivity to host cells. Tachyzoites of RH strain were incubated in NET-enriched supernatants pretreated or not with DNase for 30 min. Parasites were further used to infect Vero cells (1:1 parasite:Vero ratio) for 180 min, and the infection index was determined after 21 h culture. Data from two independent experiments are shown as mean (SD). * *p* < 0.05.

**Figure 6 microorganisms-08-01628-f006:**
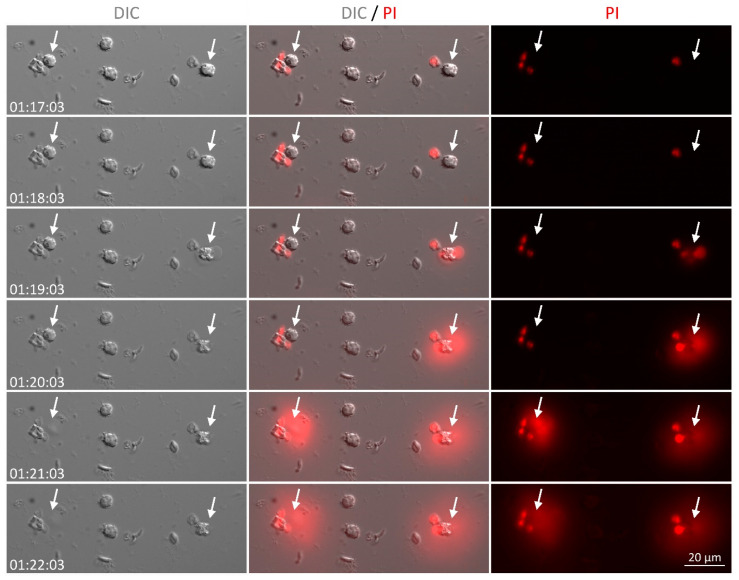
Time lapse recording of NET release by cat neutrophils in response to *T. gondii*. Neutrophils were stimulated with RH tachyzoites (5:1 parasites:neutrophil ratio) in the presence of propidium iodide (PI). Arrows point to neutrophils releasing NET after about 80 min of interaction with parasites. DIC = differential interference contrast.

**Figure 7 microorganisms-08-01628-f007:**
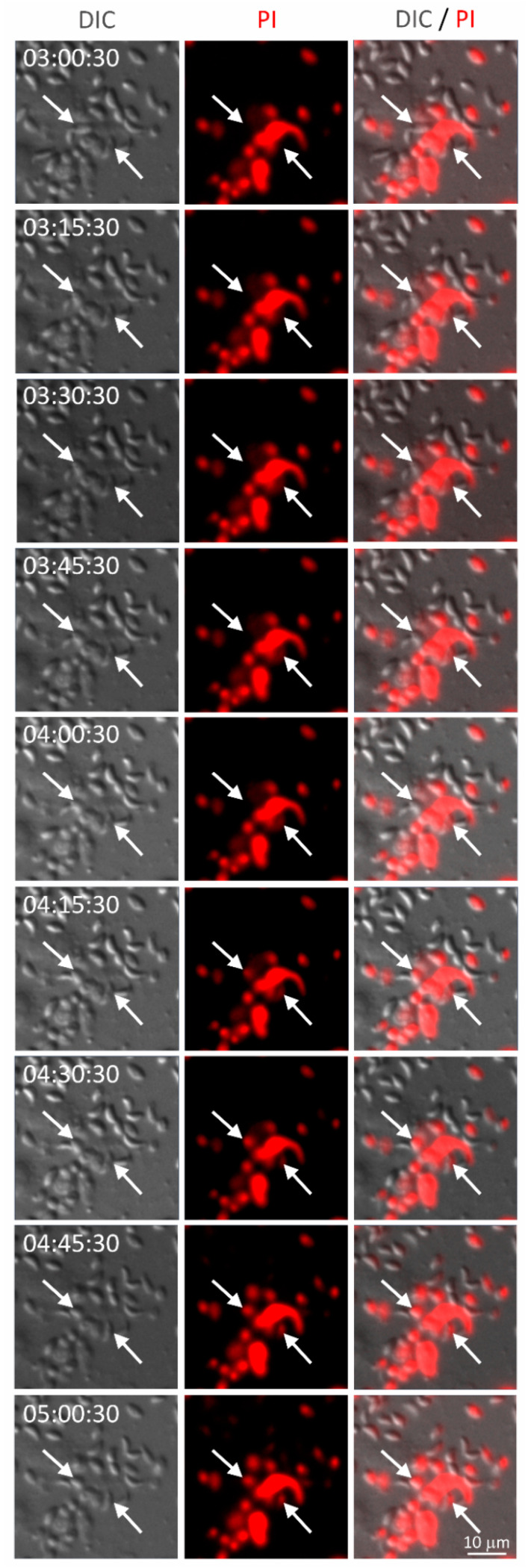
Tachyzoites of *T. gondii* die in contact with feline NET. Neutrophils were stimulated with RH tachyzoites (5:1 parasites:neutrophil ratio) in the presence of propidium iodide (PI) and images were recorded during the time. Arrows point to two parasites dying after contact with NET. DIC = differential interference contrast.

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
