# Peer review of "Extracellular Traps Released by Neutrophils from Cats are Detrimental to Toxoplasma gondii Infectivity"

_microorganisms, 2020, doi:10.3390/microorganisms8111628_

Round 1

Reviewer 1 Report

Dear Authors
Please find in the attachment my suggestions and comments

Best Regards

Reviewer 2 Report

I have some minor comments:

  1. Suggestion: remove from the title "the definitive host of the parasite"
  2. Suggestion: Keywords in alphabetical order
  3. Discussion, lines 384-390: repeated information

Reviewer 3 Report

The manuscript “Extracellular Traps Production in Response to Toxoplasma gondii by Neutrophils from Domestic Cat, the Definitive Host of the Parasite” reports that neutrophils from cat release neutrophil extracellular traps (NET) in response to 2 clonal lineages of T. gondii and that these parasites were killed by NET. These findings are of great interest in the area as they deepen in the mechanisms involved in the NET release. Overall, the manuscript is of high quality and it is suitable for the publication in Microorganisms. However, some changes must be done before being published. 

  • Line 89. Reagents. Did you use fetal bovine serum to culture Toxoplasma? Antibiotics? Please, include this information in materials and methods section.
  • Line 127. Please, could you indicate the conditions to culture Toxoplasma? (Line 136. Leishmania conditions are indicated)
  • Did you run these experiments with the other strain parasites (ME49) or only using RH strain? Lines 154, 162, 172, 178, 194. In my opinion you missed this information in materials and methods section. Please, include them.
  • Line 181: Why did you run experiments using ratio 1:1? In case it has been previously established add reference in the text.
  • Line 186. How did you estimate the total number of cells? Coverslip surface or per field of view?
  • Were experiments run in triplicates? Did you repeat them in independent manner?
  • Figure 2B. Please, improve figure quality.
  • Supplementary Figure 3. How do authors explain viability values over 100% (160% or even higher)?
  • Line 329. Did authors find that neutrophils can release NET from as early as 15 min o as early as 80 min of interaction with parasites? In figure 6 you mentioned this occurred after 80 min of interaction.
  • Figures 6 and 7. Please, could you make these figures including: DIC, PI and DIC/PI? This would help readers to make observations. With DIC/PI it is hard to observe cells and parasites.
  • Line 402. Please, could you include these data in the figure.
  • Line 445. Please, how could you explain the massive neutrophils death after treatment with DPI? Did you test different concentrations? Could you try the treatment with another molecule to study the role of NADPH oxidase and ROS production in the process of NET release? Have you thought about that?

Round 2

Reviewer 1 Report

Dear authors

I don´t have any comments other than congratulating you for your work. All my concerns were solved.

Stay Healthy